# Enhancement of Versatile Extracellular Cellulolytic and Hemicellulolytic Enzyme Productions by *Lactobacillus plantarum* RI 11 Isolated from Malaysian Food Using Renewable Natural Polymers

**DOI:** 10.3390/molecules25112607

**Published:** 2020-06-03

**Authors:** Nursyafiqah A. Mohamad Zabidi, Hooi Ling Foo, Teck Chwen Loh, Rosfarizan Mohamad, Raha Abdul Rahim

**Affiliations:** 1Department of Bioprocess Technology, Faculty of Biotechnology and Biomolecular Sciences, Universiti Putra Malaysia, UPM Serdang 43400, Selangor, Malaysia; syafiqah_zabidi92@yahoo.com (N.A.M.Z.); farizan@upm.edu.my (R.M.); 2Institute of Bioscience, Universiti Putra Malaysia, UPM Serdang 43400, Selangor, Malaysia; raha@upm.edu.my; 3Department of Animal Science, Faculty of Agriculture, Universiti Putra Malaysia, UPM Serdang 43400, Selangor, Malaysia; 4Institute of Tropical Agriculture and Food Security, Universiti Putra Malaysia, UPM Serdang 43400, Selangor, Malaysia; 5Institute of Tropical Forestry and Forest Products, Universiti Putra Malaysia, UPM Serdang 43400, Selangor, Malaysia; 6Department of Cell and Molecular Biology, Faculty of Biotechnology and Biomolecular Sciences, Universiti Putra Malaysia, UPM Serdang 43400, Selangor, Malaysia; 7Office of Vice Chancellor, Universiti Teknikal Malaysia Melaka, Jalan Hang Tuah Jaya, Durian Tunggal 76100, Melaka, Malaysia

**Keywords:** lactic acid bacteria, probiotic, *Lactobacillus plantarum*, media enhancement, cellulolytic enzyme, hemicellulolytic enzyme, lignocellulosic biomass

## Abstract

*Lactobacillus plantarum* RI 11 was reported recently to be a potential lignocellulosic biomass degrader since it has the capability of producing versatile extracellular cellulolytic and hemicellulolytic enzymes. Thus, this study was conducted to evaluate further the effects of various renewable natural polymers on the growth and production of extracellular cellulolytic and hemicellulolytic enzymes by this novel isolate. Basal medium supplemented with molasses and yeast extract produced the highest cell biomass (log 10.51 CFU/mL) and extracellular endoglucanase (11.70 µg/min/mg), exoglucanase (9.99 µg/min/mg), β-glucosidase (10.43 nmol/min/mg), and mannanase (8.03 µg/min/mg), respectively. Subsequently, a statistical optimization approach was employed for the enhancement of cell biomass, and cellulolytic and hemicellulolytic enzyme productions. Basal medium that supplemented with glucose, molasses and soybean pulp (F5 medium) or with rice straw, yeast extract and soybean pulp (F6 medium) produced the highest cell population of log 11.76 CFU/mL, respectively. However, formulated F12 medium supplemented with glucose, molasses and palm kernel cake enhanced extracellular endoglucanase (4 folds), exoglucanase (2.6 folds) and mannanase (2.6 folds) specific activities significantly, indicating that the F12 medium could induce the highest production of extracellular cellulolytic and hemicellulolytic enzymes concomitantly. In conclusion, *L. plantarum* RI 11 is a promising and versatile bio-transformation agent for lignocellulolytic biomass.

## 1. Introduction

Bacteria utilized nutrients found in the environment for its survival and cellular biosynthesis [1]. The nutritional requirements of bacteria can be designed specifically for each strain to optimize its growth in the laboratory [2,3]. As for some heterotrophs, organic carbon such as sugars, amino acids and fats are crucial for their growth [4]. The presence of free amino acids or glucose alone can generate energy essential for bacterial growth [5,6]. Enzyme-producing microbes depend on glucose as their main carbon source, while nitrogen sources of tryptone and peptone support well the bacteria growth [7]. The statistical optimization method can be employed to overcome the limitation of conventional optimization method of growth medium, whereby the statistical optimization method is able to describe the interactions between multiple variables and determine the true optimum [8,9,10]. Fractional Factorial Design (FFD) [11] is a popular first-order design of a statistical optimization approach, which is regularly employed for optimization of a culture condition.

Copious amounts of lignocellulosic biomass are expelled worldwide annually and hence they are the most abundant renewable natural polymers [12,13,14] that has vast potential to be used as nutrients for bacteria cultivation and production of biochemicals. Lignocellulose comprises cellulose, hemicellulose and lignin as complex polymers [15,16,17,18,19]. Lignin is the most recalcitrant substance in lignocellulose biomass and contributes to the physical strength of the plant cell wall, whereas cellulose and hemicellulose play an important role in cell wall construction [12]. The intra- and inter-molecular hydrogen bond of cellulose contribute to its intricate structure with a high degree of polymerization [20]. Effective degradation of lignocellulosic biomass requires the presence of both cellulolytic and hemicellulolytic enzymes.

Cellulolytic enzymes are produced by a myriad of bacteria and fungi, aerobes or anaerobes, mesophiles or thermophiles when they grow on lignocellulosic materials [21], which can be employed as a biotransformation agent in the degradation of agricultural biomasses. Two cellulolytic enzyme systems of microorganisms have been reported, which are free cellulolytic and hemicellulolytic enzymes and a protein scaffold that is known as cellulosome [22,23]. The production of cellulolytic enzymes by the producer microorganism can be induced by specific substrates [24], such as lignocellulosic biomass [25]. Despite extensive reports on the production of lignocellulosic enzymes by certain fungal and bacterial species, the efficiency of biotransformation of agro-industrial waste remains low [25], whereby phenolic compounds generated during pre-treatment of biomass have been reported to hinder cellulase and hemicellulase activities [26,27]. Moreover, a consortium of bacterial or fungal cultures are unable to achieve complete biotransformation of agro-wastes due to antagonistic interactions among the consortium members [28]. Hence, continuous efforts of screening and isolation of novel extracellular cellulase and hemicellulase producers are important to overcome these challenges.

The safety status of enzyme producers is a great concern [17] in medical, food and animal feed industries. LAB are perceived as Generally Regarded as Safe (GRAS) microorganisms by the Food and Drug Administration [29] of the United States of America since they are generally non-toxicogenic and non-pathogenic bacteria [29,30,31,32,33], which have been present in various fermented foods for decades and do not have any deleterious effects on human. In addition, LAB have a long history in industrial processes as food starters and biocontrol agents, and as producers of high-value compounds [34,35]. *Lactobacillus plantarum,* one of the major and important species of LAB [36], are notable as probiotics since they enhance the gastrointestinal health and performances in animals by promoting the gut immune response [35,36,37,38,39,40]. Furthermore, the extracellular metabolites of *L. plantarum* that are known as postbiotics have been proven extensively to be a potential alternative for in-feed antibiotic growth promoters to improve meat quality and growth performance of broiler chicken, laying hens and piglets [38,39,40,41,42]. Recently, several strains of *L. plantarum* have been reported to be versatile extracellular proteolytic enzyme producers and capable of producing an array of essential amino acids [39,43].

It is widely known that LAB are fastidious microorganisms, which required complex media compositions for their growth [44,45] and hence they could not ferment inexpensive lignocellulose feedstocks directly, whereby fermentation of lignocellulose hydrolysates by LAB has been frequently reported and is the most developed technology [29,46]. In addition, LAB are known to have the capability to degrade pentoses and hexoses via the phosphoketolase pathway [42,44].

Cellulolytic and hemicellulolytic enzymes are important industrial enzymes that are widely used in the production of food, animal feed, pulp, paper and textile. Prior to 2019, LAB have not been reported elsewhere as efficient non-treated complex biomass bioconversion agents that have the capability of producing extracellular cellulolytic and hemicellulolytic enzymes concomitantly. Thus, we have made an attempt to explore the potential of locally isolated LAB as an efficient non-treated complex biomass bioconversion agent using palm kernel cake (PKC) as a model for agricultural lignocellulosic biomass, and we have reported very recently that *Lactobacillus plantarum* RI 11, a probiotic isolated from Malaysian food, is a potential lignocellulosic biomass degrader, whereby versatile extracellular cellulolytic and hemicellulolytic enzymes were produced simultaneously when it was grown on commercially available de Man, Rogosa and Sharpe (MRS) medium [46]. However, limited knowledge regarding the effects of various non-treated complex renewable natural polymers on the growth, cellulolytic, and hemicellulolytic enzyme productions was available for LAB. Therefore, this study was conducted to investigate further the effects of various combinations of renewable natural polymers derived from agricultural wastes on the growth and productions of extracellular cellulolytic and hemicellulolytic enzymes by *L*. *plantarum* RI 11. Subsequently, the statistical optimization method of FFD was employed for further enhancement of cellulolytic and hemicellulolytic enzyme productions by *L*. *plantarum* RI 11.

## 2. Results and Discussion

### 2.1. Growth Profile of L. plantarum RI 11 in Media Supplemented with Renewable Natural Polymers

In this study, the effects of various renewable natural polymers of agricultural biomasses, such as molasses, rice straw, PKC and soybean pulp on the viability of *L. plantarum* RI 11 were determined for 7 days of incubation. Agricultural residues are the major source of lignocellulosic biomass, which is renewable, highly available and inexpensive [15,16] as rich resources of fermentable sugars [17] that can act as a precursor or intermediates in the production of biofuels and bio-based products [18]. It is a prevalent concern to the countries that have profited from agricultural industry as the disposal of biomass waste is a great challenge in the management of biomass waste [17], whereby the bioconversion of agricultural lignocellulosic biomasses remain economically unfeasible due to unavailable of efficient biocatalysts or bioagents.

In this experiment, glucose and yeast extract were used as a control for carbon and nitrogen sources, respectively [3]. Moreover, the composition of commercially available selective MRS medium was used as a reference to formulate growth medium comprising various renewable natural polymers. Figure 1a–g show the cell population of *L. plantarum* RI 11 grown in different medium formulations collected at different incubation times, whereby different renewable natural polymers that affected the growth of *L. plantarum* RI 11 substantially attributed to different nutrients were provided by different renewable natural polymers [1,2,3]. The initial cell population of approximately log 8 CFU/mL was observed for all formulated media as shown in Figure 1a–g.

Figure 1a shows the growth profile of *L. plantarum* RI 11 in control MRS medium. The maximum cell population of approximately log 11.5 CFU/mL was detected at 18 h of incubation. It was the highest cell population of *L. plantarum* RI 11 that noted in this experiment, indicating that *L. plantarum* RI 11 grown well in MRS medium since it is a selective medium for *L. plantarum.* However, the cell population decreased gradually after 1-day of incubation to log 8.5 CFU/mL at 5 days of incubation and maintained its cell population for approximately log 8.7 CFU/mL at the end of day 7 of incubation.

Figure 1b,d,g show similar growth profiles of *L. plantarum* RI 11 in media supplemented with rice straw and yeast extract (M1), molasses and yeast extract (M3), and PKC and soybean pulp (M6), respectively. The cell population of *L. plantarum* RI 11 was increased slightly from log 7.98 ± 0.12 to log 8.72 ± 0.03 CFU/mL at 12 h of incubation, log 8.04 ± 0.09 to log 9.76 ± 0.01 at 16-h of incubation and log 8.36 ± 0.05 to log 9.70 ± 0.02 CFU/mL at 12-h of incubation when grown in M1, M3 and M6 media, respectively, suggesting that the highest cell population was obtained from 12-h to 16-h of incubation for M1, M3 and M6 media, respectively. The cell population was maintained at log 8.3 CFU/mL, log 9.5 CFU/mL and log 9.4 CFU/mL for M1, M3 and M6 media, respectively, from 18-h to 22-h of incubation. In comparison, the steepest log phase of *L. plantarum* RI 11 was observed for M3 medium (Figure 1d), followed by M6 (Figure 1g) and M1 (Figure 1b) medium, respectively. M3 medium provided readily used nutrients of molasses and yeast extract as soluble carbon and nitrogen sources for the growth of *L. plantarum* RI 11. Hence, the cell population of *L. plantarum* RI 11 in M3 was higher than M1 and M6 media. Molasses is a by-product of the sugar processing industry, which is rich in reducing sugars, nitrogen, trace elements and vitamins [47,48,49,50,51] that can promote the growth of bacteria [52]. Refined sugars, such as glucose or sucrose have been used more commonly as a carbon source in comparison to rice straw and PKC [47,53], whereas molasses is commonly used fermentation substrates since it is cheap, widely available and comprised mainly sugars that act as nutrients for all microorganisms.

M6 medium that consisted of PKC and soybean pulp as complex carbon and nitrogen sources produced a steeper log phase as compared to M1 medium that consisted of rice straw and yeast extract as the complex carbon and soluble nitrogen sources. Soybean pulp is a cheap and good dietary fibre [47,48], which has been reported to encourage the growth of beneficial bacteria, such as *Lactobacillus* spp. [54]. Nevertheless, rice straw is highly heterogenous as compared to the combination of PKC and soybean pulp to support the growth of *L. plantarum* RI 11 [55]. Due to the abundance of glucose, sucrose and fructose [49,50,51,52,53], the cell population of *L. plantarum* RI 11 in M3 medium was the highest as reported by a previous study of Zajšek et al. [56].

In comparison, Figure 1c illustrates a shorter exponential phase of *L. plantarum* RI 11 when grown in M2 medium that contained complex carbon and nitrogen sources of rice straw and soybean pulp. The initial cell population of log 8.17 ± 0.04 was increased to log 10.41 ± 0.05 CFU/mL at day-2 of incubation, whereby the cell population was then maintained throughout 7-days of incubation. Despite both M2 and M6 media containing the same nitrogen source of soybean pulp, they contained heterogenous rice straw and PKC as carbon source, respectively, indicating that limited nutrients were available from the degradation of rice straw for the growth of *L. plantarum* RI 11 in comparison to PKC and thus produced shorter and steeper log phase as shown in Figure 1c.

Figure 1e shows the growth curve of *L. plantarum* RI 11 in M4 medium, which has a similar log phase to control MRS medium. The cell population was increased from initial log 8.06 CFU/mL to log 10.44 CFU/mL at day 1 of incubation. The cell population was then maintained for a day before it decreased to log 9 CFU/mL. M4 medium consisted of molasses and soybean pulp as carbon and nitrogen sources, respectively. The cell population of control MRS medium supported slightly higher cell population (log 10.67 ± 0.02) of *L. plantarum* RI 11 at 24 h of incubation as compared to M4 medium with log 10.44 ± 0.08 CFU/mL, suggesting that M4 medium has contributed a comparable effect to the growth of *L. plantarum* RI 11, whereby both molasses and soybean pulp promoted the growth of *L. plantarum* RI 11 due to their high contents of sugar and protein [47,48,49,50,51,52,53].

Figure 1f, in contrast, shows the longest and gradual slope of exponential growth of *L. plantarum* RI 11 supported by M5 medium that contained PKC and yeast extract as carbon and nitrogen sources, respectively. The initial cell population of log 8.4 CFU/mL was increased to log 9.54 CFU/mL that occurred at 16-h of incubation. Interestingly, the stationary phase occurred at 48 h of incubation, indicating that PKC was more complex substrate that could support longer growth duration of *L. plantarum* RI 11. Similar observation was also noted for medium M6 (Figure 1g) that comprised PKC and soybean pulp as the complex carbon and nitrogen sources.

Interestingly, when any carbon source combined with soybean pulp, a better cell population and shorter log phase of *L. plantarum* RI 11 was noted. This could be attributed to soybean pulp that contains a substantial amount of galactose, mannose, arabinose and uronic acid, which served as the growth promoter and enzyme inducers [47,48] for *L. plantarum* RI 11. Although soybean pulp contains mainly dietary fibre (49.85%) and lignin (29.50%), it also contains 19.79% of hemicellulose and low amount of cellulose (0.56%). The proximate analysis (results not shown) showed that the total carbon content of soybean pulp that used in this study was 71.53%. Generally, the results of Figure 1 implied that molasses was the most favorable renewable natural polymer as carbon source, followed by PKC and rice straw for the growth of *L. plantarum* RI 11. In addition, the combination of molasses and soybean pulp as soluble carbon and complex nitrogen sources promoted and sustained the growth of *L. plantarum* RI 11 that was comparable to the control MRS medium. Thus, the results obtained in this study revealed the versatility of *L. plantarum* RI 11 that possessed the ability to alter its physiological response and entered the stationary phase [57,58] accordingly when it was present in growth medium supplemented with different combinations of carbon and nitrogen sources.

### 2.2. Extracellular Cellulolytic and Hemicellulolytic Enzyme Activities of L. plantarum RI 11

Extracellular cellulolytic (endoglucanase, exoglucanase and, β-glucosidase) and hemicellulolytic (mannanase) enzyme activities of *L. plantarum* R1 11 that grown in different media mixtures were subsequently determined in this study. The cellulolytic and hemicellulolytic enzymes are essential to be present concomitantly to degrade the complex agriculture lignocellulosic biomass effectively [59], whereby endoglucanase, exoglucanase and β-glucosidase hydrolyze the glycosidic bonds [15] that are present in the cellulose polymer synergistically [19]. Endoglucanase will imitate the hydrolysis of β-1,4-glycosidic bonds of cellulose polymer randomly to produce reducing and non-reducing ends [59,60,61] of shorter cellulose polymers. Then, exoglucanase will degrade both reducing and non-reducing ends of the shorter cellulose polymer [59,60,61] to release glucose monomer. β-glucosidase will degrade cellobiose that produced by endo- and exoglucanase to 2 glucose monomers [61,62]. Endoglucanase is also known as CMCase due to its ability to hydrolyze carboxymethylcellulose (CMC), which was used as a substrate in the current study, whereas exoglucanase activity was determined by using avicel as a substrate and hence it is also known as avicelase, whereby avicel was degraded into cellobiose by exoglucanase. As for β-glucosidase activity determination, 4-nitrophenyl-β-D-glucopyranoside (PNPG) was used as a substrate in this study, whereas locust bean gum (LBG) was used as a substrate for mannanase activity determination [63,64,65]. The reducing sugar that released by both cellulolytic and hemicellulolytic enzymes was detected spectrophotometrically by using a dinitrosalicyclic acid (DNS) reagent [64,65].

Figure 2a–c demonstrate the specific endoglucanase, exoglucanase and β-glucosidase enzyme activities detected at pH 5, 6.5 and 8, respectively, that grown in the MRS control medium. It was clearly shown that *L. plantarum* RI 11 was able to produce versatile extracellular cellulolytic and hemicellulolytic enzymes that active from pH 5 to 8. Interestingly, the hemicellulolytic (mannanase) activity was only detected at pH 8 (Figure 2c) in comparison to the cellulolytic enzyme activities.

Endoglucanase-CMCase, exoglucanase-avicelase and β-glucosidase activities were detected at a broad pH range of pH 5, 6.5 and 8, respectively, as shown in Figure 2a–c, respectively, whereby three isozymes of endoglucanase were detected at pH 5 between 4–14 h of incubation (Figure 2a), one isozyme of endoglucanase was detected at pH 6.5 between 8–12 h of incubation (Figure 2b) and pH 8 at 12–14 h of incubation (Figure 2c). In comparison, two isozymes of exoglucanase were detected at pH 5 (Figure 2a) and pH 6.5 (Figure 2b) at 6–14 h and 6–12 h of incubation, respectively. Only one isozyme of exoglucanase was detected at pH 8 between 6–10 h of incubation (Figure 2c). As for β-glucosidase activity, only one isozyme was detected at pH 5, 6.5 and 8 at 12 h of incubation. However, β-glucosidase activity at pH 8 was approximately half of the activity that obtained at pH 5 and 6.5, respectively. The concomitant presence of versatile extracellular cellulolytic (endoglucanase-CMCase, exoglucanase-avicelase and β-glucosidase) and hemicellulolytic (mannanase) enzyme activities of *L. plantarum* that active at a broad pH range (from pH 5 to pH 8) corresponded well to the finding of Lee et al. [46], who has reported that LAB strains that isolated from Malaysian foods produced multi extracellular cellulolytic and hemicellulolytic enzymes simultaneously, which are active under broad pH conditions when grown in selective MRS medium. Similar observations were also reported by Nidetzky et al. [66], Eriksson [67] and Lin et al. [68]. Furthermore, Lee et al. [46] reported that β-glucosidase specific enzyme activity was the lowest among the detected cellulolytic enzymes, in which the same results were noted in this study. β-glucosidase gene is commonly harbored by LAB, especially *L. plantarum* species [69]. Hence, β-glucosidase specific activity was detected even in the control MRS medium [70]. Throughout this experiment, β-glucosidase was observed to be produced later as compared to the other two cellulolytic enzymes, as reported by Lee et al. [46].

Figure 3a–c illustrate the specific extracellular cellulolytic and hemicellulolytic enzyme activities at pH 5, 6.5 and 8, respectively, when *L. plantarum* R1 11 was grown in basal medium supplemented with rice straw and yeast extract as carbon and nitrogen sources (M1).

Surprisingly, significant (*p* < 0.05) levels of endoglucanase, exoglucanase and β-glucosidase activities were detected at pH 5, 6.5 and 8, respectively, as found in MRS control medium. One isozyme of endoglucanase was recorded at pH 5 and 6.5, respectively, both at 10-h of incubation with 4.07 µg/min/mg and 1.68 µg/min/mg of specific activities. As for exoglucanase, one isozyme was detected at pH 5 between 8–12 h of incubation, whereas two isozymes were detected at pH 6.5 between 4 to 10-h of incubation. For β-glucosidase activity, only one isozyme was detected at pH 5, 6.5 and 8 after 12-h of incubation, whereby the specific activity was reduced to half as compared to the β-glucosidase activity that obtained in MRS control medium. The specific enzyme activities recorded for β-glucosidase were 2.54 nmol/min/mg and 2.10 nmol/min/mg at pH 5 and 6.5, respectively. As for pH 8, a significant (*p*<0.05) level of β-glucosidase was detected, which was 6.05 nmol/min/mg. No mannanase activity was detected when *L. plantarum* R1 11 was grown in M1 medium. The cellulolytic and hemicellulolytic enzyme levels were comparably low in M1 medium. Hence, it can be concluded that though yeast extract was able to promote the growth of *L. plantarum* R1 11, but, due to the heterogeneity of the rice straw [55], a low level of cellulolytic and hemicellulolytic enzymes was secreted by the *L. plantarum* R1 11.

Figure 4a–c demonstrate the significantly (*p*<0.05) lower specific extracellular endoglucanase, exoglucanase and β-glucosidase activities at pH 5, 6.5 and 8, respectively, when *L. plantarum* R1 11 was cultivated in basal medium supplemented with rice straw and soybean pulp (M2) medium. Interestingly, only one isozyme of extracellular endoglucanase was detected at pH 5, 6.5 and 8, respectively, between 4–6 h of incubation with significantly (*p*<0.05) higher activity of 3.1 µg/min/mg detected at pH 5. However, this specific activity was significantly lower (*p* < 0.05) than those specific activity observed for other media. Similar levels of enzymatic activities were observed in both Figure 3 and Figure 4. This was most probably due to the same carbon source used in both media. Carbon source is very crucial in the production of enzyme [71]; hence, it can be concluded that rice straw could not promote the production of cellulolytic and hemicellulolytic enzymes. As for the exoglucanase specific activity, only one isozyme was detected at pH 5 and 6.5 between 4–6 h and between 8–10 h of incubation respectively. As for β-glucosidase, it was detected at pH 5 and 6.5. However, significantly higher (*p*<0.05) specific activity of 4.1 nmol/min/mg at 16 h of incubation was detected at pH 5. In comparison, much lower mannanase activity was recorded between 1.2 to 1.5 µg/min/mg at 14-16 h of incubation at pH 8. Generally, insignificant (*p*>0.05) level of cellulolytic and hemicellulolytic enzyme specific activities were obtained when *L. plantarum* R1 11 was grown in basal medium supplemented with rice straw and soybean pulp, attributing to the recalcitrant nature of rice straw [55].

Figure 5a–c show the specific extracellular endoglucanase, exoglucanase and β-glucosidase activities detected at pH 5, 6.5 and 8, respectively, when *L. plantarum* R1 11 was grown in basal medium supplemented with molasses and yeast extract (M3). At pH 5, two isozymes of endoglucanase were detected between 4–12 h of incubation. However, only one isozyme of endoglucanase was detected at pH 6.5 and 8, respectively, between 4–10 h of incubation. Simultaneously, exoglucanase specific activity was also recorded with endoglucanase activity from pH 5 to 8, despite only one isozyme was detected at pH 5 between 2–6 h of incubation. Nevertheless, two isozymes of exoglucanase were detected at pH 6.5 and 8, respectively, between 4–16 h of incubation. In comparison, the highest specific activity of β-glucosidase was only produced after endoglucanase and exoglucanase were excreted by *L. plantarum* R1 11, which occurred between 10–16 h of incubation. Interestingly, three significant (*p*<0.05) enzymatic activities of mannanase were detected at pH 8 between 6–16 h of incubation (Figure 5c).

The finding in this experiment was contradicted with a study reported by Kim et al. [72], whereby the present of mannan-like polysaccharide was essential to induce the production of mannanase. However, mannan-like polysaccharide was not present in the M3 medium that contained molasses as carbon source. Nevertheless, El-Sharouny et al. [73] reported that the production of mannanase was affected by glucose. Molasses, as reported by Hamouda et al. [53], consists of sucrose, glucose, fructose, xylose and maltose, hence it was proven that molasses can induce the production of mannanase. Furthermore, the mannanase enzyme that purified by El-Sharouny et al. [73] was active in extreme alkaline conditions, which was similar to the observation of this experiment.

Molasses is acidic in nature, in which it provides a favorable environment for LAB to grow and to produce useful metabolite [49,50,51,52,53]. Molasses is also cellulosic [69], in which it will act an inducer for the production of cellulolytic enzymes by cellulolytic microorganisms [45] before 20-h of incubation [74] as shown in this experiment, whereby M3 medium was able to induce the production of three cellulolytic enzymes over a broad pH range by *L. plantarum* R1 11 with the highest and significant endoglucanase was noted from pH 5 to 6.5 in M3 media. In addition, M3 medium is a more economical medium that contained readily fermented sugars [50,51,52] to produce comparable extracellular cellulolytic and hemicellulolytic enzymes in comparison to the MRS control medium

Figure 6a–c demonstrate the specific endoglucanase, exoglucanase and β-glucosidase obtained at pH 5, 6.5 and 8, respectively, when *L. plantarum* R1 11 was grown in basal medium supplemented with molasses and soybean pulp (M4). Significantly (*p* < 0.05) higher endoglucanase specific activities that obtained approximately at 8 h of incubation was detected at pH 5 in comparison to pH 6.5 and pH 8, respectively, whereby only one isozyme of endoglucanase was produced in M4 formulated medium. The exoglucanase, on the other hand, recorded a higher specific activity of approximately 10–20 µg/min/mg at pH 5 and 8, respectively. Though the highest endoglucanase was detected in M4 medium, it was only active at pH 5. In contrast, M3 medium that consists of molasses and yeast extract produced more stable endoglucanase enzymes throughout the broad pH range. As for the β-glucosidase activity, low specific activity of approximately 3 nmol/min/mg was obtained between 4–14 h of incubation from pH 5 to pH 8, despite of insignificant (*p* > 0.05) activity, was noted. Interestingly, mannanase activity was detected at pH 6.5 and pH 8 between 12–18 h and 8–12 h of incubation, respectively. However, the highest specific activity of 8.05 µg/min/mg was noted at pH 6.5. This finding was similar to the study of Kim et al. [72], in which mannan-like polysaccharides from soybean pulp induced mannanase productions. Interestingly, mannanase was produced concomitantly with the production of endoglucanase, as reported by Sachslesner et al. [75]. Hence, it was generally observed that, throughout this experiment, mannanase enzyme was detected together with endoglucanase activity. However, not all endoglucanase enzyme productions were accompanied by the production of mannanase enzyme.

Figure 7a–c show the specific endoglucanase, exoglucanase and β-glucosidase activities which were detected at pH 5, 6.5 and 8, respectively, when *L. plantarum* R1 11 was grown in basal medium supplemented with PKC and yeast extract (M5). For this formulation, one isozyme of endoglucanase was present at pH 6.5 and 8 respectively with significantly lowest (*p* < 0.05) activities of 1–2.7 µg/min/mg detected. As for exoglucanase activity, although it was detected from pH 5–8, an insignificant (*p* > 0.05) level of activity was recorded. Interestingly, significant specific activities of β-glucosidase were detected between 4–8 h of incubation, having 7.55 µg/min/mg at pH 8 as the highest activity. Surprisingly, no mannanase was detected from M5 medium. This finding was in contrast with the study reported by Kim et al. [72], whereby PKC that comprised of mannan polysaccharides did not induce the mannanase production in this study. Nevertheless, the results of this experiment were in close agreement with the finding of El-Sharouny et al. [73], whereby the production of mannanase was induced by the presence of simple sugars such as glucose.

Figure 8a–c illustrate the specific endoglucanase, exoglucanase and β-glucosidase activities recorded at pH 5, 6.5 and 8, respectively, when *L. plantarum* R1 11 was grown in basal medium supplemented with PKC and soybean pulp (M6). Surprisingly, extracellular endoglucanase activity was not detected at all pH and only one isozyme of exoglucanase was detected at pH 5 with an insignificant (*p*<0.05) activity of 2.56 µg/min/mg was recorded. Similarly, although β-glucosidase was noted from pH 5 to 8, the specific enzyme activity was diminutive. Mannanase, as usual, was detected only at pH 8 with low level of specific enzyme activity at a later incubation time of 16–18 h. The basal medium supplemented with PKC and soybean pulp did not induce substantial production of extracellular cellulolytic and hemicellulolytic enzymes by *L. plantarum* RI 11 in comparison to the other renewable natural polymers such as molasses and rise straw as shown clearly in Figure 3, Figure 5 and Figure 6, respectively, indicating that the combination of PKC and soybean pulp was not a favorable medium for the production of extracellular cellulolytic and hemicellulolytic enzymes by *L. plantarum* RI 11. However, mannanase was detected later throughout the incubation, though the enzymatic level was very low. This finding coincided with the study of El-Sharouny et al. [73], as well as noted in Figure 7 of this study.

Generally, the overall results of this experiment showed that the basal medium supplemented with molasses and yeast extract (M3) recorded as the best basal medium that was comparable with the MRS control medium to produce extracellular cellulolytic and hemicellulolytic enzymes, respectively. Cellulolytic enzyme is a complex enzyme comprising endoglucanase, exoglucanase and β-glucosidase [75,76], whereas hemicellulolytic enzyme encompasses xylanase and mannanase [77]. The extracellular cellulolytic and hemicellulolytic enzyme activities of *L. plantarum* RI 11 were detected within 24 h of incubation, and hence prolong incubation time resulting in no further extracellular enzyme production despite the cell viability that could maintain up to seven days of incubation at 30 °C as shown in Figure 1. This could be due to carbon starvation [57] and led to a halt of the protein and DNA synthesis and hence the enzyme production. Interestingly, β-glucosidase specific activity was only detected after the production of both endoglucanase and exoglucanase specific activities, supporting the synergistic enzymatic degradation mechanism of cellulosic polymer, whereby β-glucosidase cleaved the cellobiose molecules that released from the enzymatic reaction of both endoglucanase and exoglucanase [15,17] into monomer glucose [23,60]. Generally, β-glucosidase activity is the rate limiting factor among the cellulolytic enzymes, which is very sensitive to glucose inhibition [70,77]. However, the glucose and xylose contents of molasses are good inducers that could stimulate bacteria producer to synthesize β-glucosidase. Nonetheless, among the cellulolytic enzymes, the specific enzyme activity of β-glucosidase was the lowest, as reported by Lee et al. [46]. Moreover, Spano et al. [69] reported that β-glucosidase gene isolated from wine *L. plantarum* was regulated by abiotic stresses.

The results obtained in this study also indicated that the extracellular hemicellulolytic enzyme of mannanase was significantly (*p* > 0.05) produced by *L. plantarum* RI 11 between 6–16 h of incubation (3.96 to 8.09 µg/min/mg) that active at alkaline pH 8, as reported by Rasul et al. [74]. Furthermore, mannanase and endoglucanase were usually produced concomitantly and production of mannanase was always accompanied by the production of endoglucanase [75]. Mannanase of *L. plantarum* RI 11 was most likely to be induced by the presence of mannose or mannan-like polysaccharides [72] or sugar [73] in the formulated media. Mannanase is essential to cleave mannan to mannose that could serve as energy source for the survival of bacteria producer cells [69,70]. However, since lower specific enzyme activities of β-glucosidase and mannanase were secreted by the *L. plantarum* RI 11, it was possible that the cellulose degradation by endoglucanase and exoglucanase was inhibited by the presence of cellobiose [23]. Hence, both β-glucosidase and mannanase enzymes were essential to warrant complete degradation of renewable natural polymers [71,72,73,74].

### 2.3. Fractional Factorial Design for Growth Enhancement of L. plantarum RI 11

Factorial Design is a popular statistical optimization method to determine important factors that influence the yield of experiments and the interactions that might exist between the factors [78,79,80]. Full factorial design is more suitable for factors less than 4. Fractional Factorial Design (FFD) is more appropriate when the resources are limited [79] and there are more than five investigated factors [79,80]. It is important to know the effect of medium components on the growth of *L. plantarum* RI 11. Therefore, the medium formulation that gave the highest viable cell count was optimized subsequently by FFD. Altogether, 16 media formulations were suggested by FFD based on carbon and nitrogen sources used in this study. Both control MRS medium and reconstituted control MRS (CRMRS) medium were used as control media in this experiment, with the former being commercial and the latter was reconstituted. The renewable natural polymers that used as carbon sources in this experiment were rice straw, molasses and PKC, whereas soybean pulp and yeast extract were employed as nitrogen sources in this experiment. In each formulated medium, other minerals and salts such as Tween 80, sodium acetate, potassium hydrogen phosphate, diammonium hydrogen citrate, magnesium sulphate heptahydrate and manganese sulphate tetrahydrate were added according to the composition of control MRS medium. Tween 80 and potassium hydrogen phosphate were reported to have a positive effect on the growth of microorganisms [48] since both elements can maintain pH value to prolong a fermentation process.

Table 1 shows the viable cell count of *L. plantarum* RI 11 grown in 16 formulated media, and both control MRS medium and CRMRS medium for seven days of incubation. The viable cell count of *L. plantarum* RI 11 decreased significantly (*p* < 0.05) for the first five days of incubation when grown in F1 medium (basal media with no carbon and nitrogen source) and no growth was detected after day 5 of incubation. Under unfavorable conditions, all bacteria will develop starvation-survival strategy to sustain their growth [81,82] and bacterial cell will enter a dormant state [83,84,85] when grown in basal medium without carbon and nitrogen sources. F4 medium (glucose, molasses and soybean pulp) and F5 medium (rice straw, yeast extract and soybean pulp) recorded the highest viable cell count of log 11 CFU/mL at 4-days of incubation, which was also noted at Section 2.1, whereby nitrogen source of soybean pulp [86] was sufficient to promote the growth of *L. plantarum* RI 11 [48,49] and too much of nitrogen source was reported to cause cell death [87]. Soybean pulp has been reported to increase the cell population of *Lactobacillus* sp. [86,88], whereas rice straw can also act as the nitrogen source [55]. All nitrogen sources used in this study were organic and especially yeast extract [87] in formulated F5 medium supported the rapid growth and high viable count of *L. plantarum* RI 11. Yeast extract that consists of amino acids, peptides, vitamins and carbohydrates [87] is a good nitrogen source to stimulate metabolic stress [88]. In addition, yeast extract also contains unidentifiable growth factors that can stimulate the growth of microorganisms [83].

The growth of *L. plantarum* RI 11 increased substantially when soluble sugars such as glucose or molasses was combined with soybean pulp as the organic nitrogen source [83,85]. Amino acids were made available upon the degradation of nitrogen source [83]. In addition, media containing glucose and disaccharides of glucose would decrease the pH of the media [83], which was more favorable for the growth of *L. plantarum* RI 11. Different carbon sources will enter the metabolic network at different entry points of glycolysis [89]. Nerveless, the bacteria will usually use preferred substrates or substrates that give a higher growth rate [90,91]. For an example, F4 medium that contained both glucose and molasses that function as carbon source, *L. plantarum* RI 11 has to choose either glucose or molasses as its sole carbon source. Selection of the carbon sources is made at the level of carbohydrate-specific induction [83] known as carbon catabolite repression [90,91]. For instance, the presence of sucrose in molasses will promote the high biomass yield due to the presence of intermediates in the metabolism of fructose. Too many carbon and nitrogen sources as found in F15 and F16 media would increase the cell density until a maximum point in which it would decrease subsequently, due to inhibitory effects of the substrates [90,92]. Between soluble and complex carbon sources, cellulose-producing microorganisms will usually select complex carbon sources, as reported by Dou et al. [93].

A similar level of cell densities was detected for both F4 and F5 media, although both media sources have different compositions. It could be due to both yeast extract and soybean pulp, in high concentration, acting as carbon sources as well as nitrogen sources [90]. F4 and F5 media contained less substrates when compared to F15 and F16 media, which contributed to different cell populations obtained. Therefore, the results of this experiment demonstrated that the medium compositions of F4 (glucose, molasses and soybean pulp) and F5 (rice straw, soybean pulp and yeast extract) contributed comparable cell populations with control MRS medium.

### 2.4. Fractional Factorial Design for Enhancement of Cellulolytic and Hemicellulolytic Enzyme Productions

The corresponding extracellular cellulolytic (endoglucanase-CMCase, exoglucanase-avicelase and, β-glucosidase) and hemicellulolytic (mannanase) enzyme activities of *L. plantarum* R1 11 that grown in 16 different media sources formulated by FFD were determined subsequently. Figure 9 illustrated that *L. plantarum* R1 11 produced different levels of extracellular cellulolytic and hemicellulolytic enzyme activities in various formulated media sources. Surprisingly, the medium that produced high viable cell density of *L. plantarum* R1 11 in Section 2.3 did not correspond to the medium that generated high cellulolytic and hemicellulolytic specific enzyme activities in this experiment.

There was no production of cellulolytic and hemicellulolytic enzymes when *L. plantarum* R1 11 was grown in F1 medium (basal medium without any carbon and nitrogen sources). *L. plantarum* R1 

11 was most likely to be in a dormant stage, repressing its metabolic activity [82] until the favorable growth condition was available [87]. Both carbon and nitrogen sources were highly important for the growth and production of metabolites by microorganisms [94,95].

Generally, less enzymes with low level of enzymatic activities were noted when *L. plantarum* R1 11 was grown in F10, F15 and F16 formulated media. F10 was a basal medium supplemented with both soluble (glucose) and heterogeneous renewable natural polymers of PKC, yeast extract and soybean waste, indicating that the supplementation of many carbon sources in the growth medium would not necessarily enhance the production of cellulolytic and hemicellulolytic enzymes by *L. plantarum* R1 11. Glucose is a simple carbohydrate [44], which acts as a good energy source [83]. However, glucose would repress the cellulolytic and hemicellulolytic enzyme activities [83,85,93]. Nevertheless, a low endoglucanase activity (0.90 µg/min/mg) was detected at pH 5. The repression effect of PKC was greater and hence only a low amount of activity was detected, although *L. plantarum* RI 11 was able to grow on PKC [93]. F15 and F16 media that consisted PKC demonstrated low level of mannanase activity. However, a higher level of mannanase activity was detected in F16 medium. PKC has been reported to be a good substrate for mannanase production [94]. PKC, which comprises of mannan, will induce the production of mannanase [72]. F16 also contained glucose, rice straw, molasses, soybean pulp and yeast extract. Hence, PKC was a better substrate to produce mannanase by *L. plantarum* RI 11 [72,84].

Similarly, a low level of enzyme activity was detected for F5, F6, F7, F9, F11, and F13 media. Different enzyme productions were detected, but the levels of enzymatic activity were insignificant (*p*>0.05). F5 medium was a basal media supplemented with rice straw, yeast extract and soybean pulp. The positive effects of yeast extract and soybean pulp as a nitrogen source [95] have been demonstrated on the viable cell density of the microorganisms [95]. In this experiment, rice straw was acted as both carbon and nitrogen sources [84]. Hence, most of the medium components contributed to the growth of the cell, resulting in microbial biomass yield. Rice straw, which consisted of a similar proportion of cellulose and hemicellulose [96,97], induced the production of cellulolytic enzymes instead of hemicellulolytic enzymes by *L. plantarum* RI 11. Figure 9d shows similar levels of endo-, exoglucanase and β-glucosidase at different pH that produced at different incubation times. Endoglucanase specific activity that active from pH 5–8 was detected from day-1 to day-5 of incubation, whereas exoglucanase was detected at day-2 and day-3 of incubation. Moreover, β-glucosidase, on the other hand, was detected at day-2 to day-5 of incubation, which suggested strongly the synergistic cellulose degradation by *L. plantarum* RI 11 [76,96]. β-glucosidase enzyme production, which happened later than the incubation time, supported a previous report by Lee et al. [46].

F6 medium supplemented with glucose, rice straw and soybean pulp has recorded the highest viable cell population with corresponding extracellular cellulolytic and hemicellulolytic enzyme productions in comparison to F5 medium that induced only cellulolytic enzymes. This was most probably due to nutrient content of both rice straw and soybean pulp that consisted both cellulose and hemicellulose fractions [98,99]. However, soybean pulp is rich in mono- and oligosaccharides [86], which might contribute to the inhibition of cellulolytic and hemicellulolytic enzyme productions. Similar observation was noted in Figure 9h, whereby F9 medium was consisted of PKC and soybean pulp. Despite PKC being a good substrate for various hydrolytic enzyme productions [94], the presence of soybean pulp would inhibit the production of various hydrolytic enzymes [88]. Increased specific activity of all cellulolytic enzymes was observed in F7 medium, which was supplemented with molasses and rice straw. Therefore, the increased specific activity of all cellulolytic enzymes could be attributed to molasses and rice straw supplementations, which were highly heterogeneous in nature. Molasses is rich in mono- and oligosaccharides [46] and oligosaccharides also can be generated by incomplete digestion of endo- and exoglucanase enzymes [99,100]. However, since *L. plantarum* RI 11 was able to produce β-glucosidase enzymes [70], it was most likely that the oligosaccharides were converted to glucose. Hence, no apparent effect of enzyme inhibition was noted in this experiment.

Both F11 and F13 media contained PKC and yeast extract, despite a difference with the addition of molasses in the former medium, and rice straw for the latter medium. Surprisingly, endoglucanase enzyme activity was detected in both media, probably due to the presence of PKC [91]. In addition, exoglucanase was also detected in F11 medium. However, β-glucosidase was detected in F13 medium, implying that molasses promoted exoglucanase enzyme production and rice straw promoted β-glucosidase enzyme production by *L. plantarum* RI 11.

F3, F4 and F8 media was observed to promote the production of all cellulolytic and hemicellulolytic enzymes simultaneously with mannanase being recorded as the highest activity of approximately 8 U/min/mg. F3 and F4 media comprised molasses and soybean pulp with the addition of yeast extract for F3 medium and addition of glucose for F4 medium, respectively. F8 medium, on the other hand, comprised of glucose, molasses, rice straw and yeast extract. The enzyme production for both F3 and F4 media occurred at later incubation time as the enzyme production of F8 medium was occurred at early stage of the incubation. For F3 and F4 media, cellulolytic enzyme activity was detected at the early stage of incubation, then followed by the mannanase production at the later of incubation. Molasses, which is consisted mainly mono-and oligosaccharides [55,60,61,62], induced the production of endo- and exoglucanase in this experiment. In comparison, *L. plantarum* RI 11 required a longer time to fully utilize soybean pulp and hence the mannanase was produced at the later stage of incubation. A similar observation was noted for F8 medium. However, mannanase was detected at day-1 and day-3 of incubation along with other cellulolytic enzyme productions.

F12 medium recorded the highest specific activity of endo-, exoglucanase and mannanase as compared to the other formulated media. F14 medium, on the other hand, recorded the highest β-glucosidase enzyme production. Both F12 and F14 media were the best in producing the cellulolytic and hemicellulolytic enzymes concomitantly. Approximately 41 µg/min/mg of endoglucanase specific activity was detected at pH 5 and 6.5, respectively. However, the endoglucanase specific activity was reduced to half at pH 8. Similar levels of endoglucanase at pH 8 was detected for exoglucanase at pH 5. Approximately 26 µg/min/mg was recorded for exoglucanase at day-2 of incubation. F12 medium, which comprised of glucose, molasses and PKC, was a good substrate to produce these enzymes at day-2 of incubation. PKC and molasses induced the production of endoglucanase, exoglucanase and mannanase enzymes in this experiment. The mono- and oligosaccharides of molasses and PKC were favorable medium components to produce both cellulolytic and hemicellulolytic enzymes, rather than inhibiting as shown by soybean pulp [89,90]. The results obtained in this study have clearly shown that both endo- and exoglucanase activities were highly correlated with each other [98,101] and reacted synergistically for efficient agriculture biomasses degradation. Approximately, 10 µg/min/mg of mannanase was detected only at pH 8. It could be deduced that the mannanase production by *L. plantarum* RI 11 was active at pH 8, which was similar to the mannanase production by *Bacillus* sp. that was also active at pH 8 [102].

As for F14 medium, exoglucanase specific activity was reported to be similar to those produced by F12 medium that were active at a broad pH range. F14 medium that contained glucose, rice straw and PKC was a good substrate for exoglucanase, but not for endoglucanase production. A lower level of endoglucanase was detected in F14 medium, supporting a previous notion that molasses supported endoglucanase production. However, the highest β-glucosidase was detected in F14 medium at pH 5, which was in agreement with the study of carbon and nitrogen sources’ influences on the growth and sporulation of *Bacillus thuringiensis* for biopesticide production [103]. Despite sugar or monosaccharides possibly inhibiting enzyme production, it only affected other cellulolytic enzymes but not β-glucosidase [103]. Hence, higher β-glucosidase specific activity was resulted, although glucose was present in the media.

## 3. Materials and Methods

### 3.1. Microorganisms and Maintenance

*L. plantarum* RI 11 isolated from Malaysian fermented food (*ikan rebus*) was obtained from the Laboratory of Industrial Biotechnology, Department of Bioprocess Technology, Faculty of Biotechnology and Biomolecular Sciences, Universiti Putra Malaysia (UPM) [35]. The stock culture of *L. plantarum* RI 11 was maintained at −20 °C in MRS broth (Merck, Darmstadt, Germany) containing 20% (*v*/*v*) of glycerol (Merck) [46]. The reviving procedure of *L. plantarum* RI 11 was performed according to the method described by Foo et al. [35].

### 3.2. Media Formulation for L. plantarum RI 11

The active *L. plantarum* strain was washed once with sterile 0.85% (*w*/*v*) NaCl (Merck) solution and adjusted to 10^9^ CFU/mL to be used as inoculum. The adjusted inoculum of *L. plantarum* RI 11 was then grown in basal media containing 1 g L^−1^ Tween 80 (Merck), 5 g L^−1^ sodium acetate (Merck, Darmstadt, Germany), 2 g L^−1^ potassium hydrogen phosphate (Merck), 2 g L^−1^ diammonium hydrogen citrate (Merck), 0.2 g L^−1^ magnesium sulphate heptahydrate (Merck) and 0.04 g L^−1^ manganese sulphate tetrahydrate (Merck), which was supplemented with different combination of carbon and nitrogen sources. The renewable natural polymers that were used as carbon sources were rice straw (15.46 g/L), molasses (25.09 g/L) and PKC (11.86 g/L), whereas soybean pulp (51.54 g/L) and yeast extract (28.34 g/L) were used as nitrogen sources in this experiment.

Table 2 shows six media combinations of carbon and nitrogen sources supplemented in basal medium were designed in this experiment, whereas Table 3 shows the carbon and nitrogen content of each renewable natural polymers that were used for the formulation of growth media in this study. Glucose (Merck), PKC, rice straw and molasses were employed as carbon sources, whereas yeast extract and soybean pulp were used as nitrogen sources. The amount of each carbon and nitrogen source supplemented in the basal medium was determined with the reference to the carbon and nitrogen contents of commercially available MRS broth that were employed as a control medium in this study. Yeast extract was purchased from Ohly (Hamburg, Germany). Both glucose and yeast extract were employed as the control for carbon and nitrogen sources in this experiment.

### 3.3. Extracellular Cellulolytic and Hemicellulolytic Enzymes Activities of L. plantarum RI 11

Ten (10) % (*v*/*v*) of the active *L. plantarum* RI 11 culture (log 9 CFU/mL) was transferred into 9 mL of the basal medium supplemented with renewable natural polymers derived from agro wastes and incubated for 7 days at 30 °C. Samples were harvested every 2-h intervals for 7 days of incubation. The cell biomass of *L. plantarum* RI 11 was collected by centrifugation at 10,000× *g* for 15 min (Hitachi Himac CR22611 High Speed Refrigerated Centrifuge, Hitachi Koki Co Ltd., Tokyo, Japan) [46]. The cell population of *L. plantarum* RI 11 of each sampling interval was freshly analysed to determine the cell viability, whereas the supernatant was filtered by 0.2 μm cellulose acetate hydrophilic syringe filter (Minisart, Sartorius Stedim Biotech GmbH, Göttingen, Germany) to obtain cell-free supernatant (CFS). CFS was then used for the determinations of protein concentration, extracellular cellulolytic and hemicellulolytic enzymatic activities in triplicate analyses.

### 3.4. Protein Concentration Determination

The protein concentration was determined in triplicate analyses by using the Bradford reagent (Sigma, St. Louis, MO, USA) method [104] and bovine serum albumin (BSA) (Sigma) was used as a reference.

### 3.5. Cell Viability Determination

Standard plate count method [43] was conducted to determine the cell population of *L. plantarum* RI 11. The cell pellet was washed once with 0.85% (*w*/*v*) sodium chloride (NaCl) (Merck) solution. Tenfold serial dilution (10^0^ to 10^−9^) was conducted for colony forming unit/mL (CFU/mL) determination using 0.85% (*w*/*v*) NaCl solution as the diluent. An aliquot of 50 μL from the respective diluted cell population was spread evenly on the MRS agar and incubated at 30 °C for 48 h. The cell viability determination was conducted in triplicate.

### 3.6. Quantification of Extracellular Cellulolytic and Hemicellulolytic Enzyme Activities

Extracellular cellulolytic and hemicellulolytic enzyme assays were determined in triplicate with three types of assay buffers: 0.1 M sodium acetate (Merck) pH 5 buffer, 0.1 M sodium phosphate (Merck) pH 6.5 buffer and 0.1 M Tris-HCl (Merck) pH 8 buffer as described by Lee et al. [46]. The substrates used were 2.0% (*w*/*v*) of carboxymethylcellulose (Sigma), 0.5% (*w*/*v*) of avicel pH10 (Fluka, Tokyo, Japan), 5 mM 4-nitrophenyl- β-D-glucopyranoside (PNPG) (Sigma) and 1% (*w*/*v*) of locust bean gum from ceratonia silique seed (LBG) (Sigma) for endo-β-1,4-glucanase (CMCase), exo-β-1,4-glucanase (avicelase), β-glucosidase and mannanase activity, respectively [63,64,65,105]. CMCase, avicelase and β-glucosidase assays represented cellulolytic enzyme activities, while mannanase assay denoted hemicellulolytic enzyme activity. CFS collected from different medium formulation and incubation times were incubated with respective substrates prepared in three different buffers and incubated at 37 °C for 1 h. The reducing sugar was determined by using dinitrosalicyclic acid (DNS, Sigma) assay method for CMCase, avicelase and mannanase assays [63,64,65,105,106,107]. The absorbance of each sample was determined at 540 nm by Cary 50 Probe UV-visible spectrophotometer (Agilent Technologies, Santa Clara, CA, USA) [63,64,65,105].

As for β-glucosidase assay, the CFS was incubated with respective substrate prepared in three different buffers as described above at 37 °C for 30 min, followed by adding 500 μL of cold 0.5 M sodium carbonate (Sigma) to terminate the enzyme activity and the absorbance of assay mixture was determined at 373 nm [63] by Cary 50 Probe UV-visible spectrophotometer (Agilent, Santa Clara, CA, USA). The positive controls of all enzyme assays comprised CFS and respective buffers, while negative control comprised respective substrates and buffers. Glucose was utilized as the CMCase and avicelase reference. As for β-glucosidase and mannanase, p-nitrophenol and mannose were used as the references, respectively. The specific enzyme activity (U/min/mg) for CMCase, avicelase and mannanase was defined as the amount of reducing sugar produced (μg) per minute of incubation time in 1 mg of protein under experimental conditions. The enzyme activity (U/min/mg) of β-glucosidase was defined as the amount of enzyme that released 1 nmol of p-nitrophenol per minute of incubation time in 1 mg of protein under experimental conditions.

### 3.7. Fractional Factorial Design for Growth Enhancement and L. plantarum RI 11

Three replicates for each of the basal media supplemented with renewable natural polymers were prepared according to Table 4. The renewable natural polymers that used as carbon sources were rice straw (15.46 g/L), molasses (25.09 g/L) and PKC (11.86 g/L), whereas soybean pulp (51.54 g/L) and yeast extract (28.34 g/L) were used as nitrogen sources in this experiment. In each formulated medium, other minerals and salts such as Tween 80 (1.00 g/L), sodium acetate (5.00 g/L), potassium hydrogen phosphate (2.00 g/L), diammonium hydrogen citrate (2.00 g/L), magnesium sulphate heptahydrate (0.20 g/L) and manganese sulphate tetrahydrate (0.05 g/L) were added as the additional nutrients. The amount of the minerals and salts followed the compositions of control MRS medium.

### 3.8. Statistical Analysis

Analysis of variance (ANOVA) was used to determine the significant difference between results. The significance of the differences between the mean was compared by using Duncan‘s Multiple Range Test System at *p*-value less than 0.05. Statistical analyses were performed using SAS statistical software (version 9.4). The results of statistical analysis were presented as mean ± standard error of the mean (SEM).

## 4. Conclusions

The results obtained in current study suggested that renewable natural polymers such as rice straw, molasses, PKC and soybean pulp were able to induce *L. plantarum* RI11 to produce versatile extracellular cellulolytic and hemicellulolytic enzymes that active from acidic to alkaline pH conditions. Diverse levels of a few isozymes of extracellular cellulolytic and hemicellulolytic enzymes that produced within the 24 h of incubation warrant extensive research effort in the mechanism elucidation of these inducible enzymes by the renewable natural polymers. Different natural biomass substrates elicited different effects on *L. plantarum* RI11 variability and cell population, as well as the production of various extracellular cellulolytic and hemicellulolytic enzymes. The versatile extracellular cellulolytic and hemicellulolytic enzymes production by *L. plantarum* RI11 that active under broad pH range possessed vast application potential in various industries. Amongst the formulated media, F12 medium that consisted of glucose, molasses and PKC promoted the growth of *L. plantarum* RI 11 that comparable to commercially available control MRS medium together with the concomitant production of the highest activities of extracellular endoglucanase, exoglucanase, β-glucosidase and mannanase enzymes that active under broad pH conditions. Hence, *L. plantarum* RI 11 isolated from Malaysian food is a versatile cellulolytic and hemicellulolytic enzymes’ producer and as a potential bioagent for biotransformation of renewable natural polymers. This new finding warrants future explorations in the production of extracellular cellulolytic and hemicellulolytic enzymes by the non-toxigenic and non-pathogenic LAB.

## Figures and Tables

**Figure 1 molecules-25-02607-f001:**
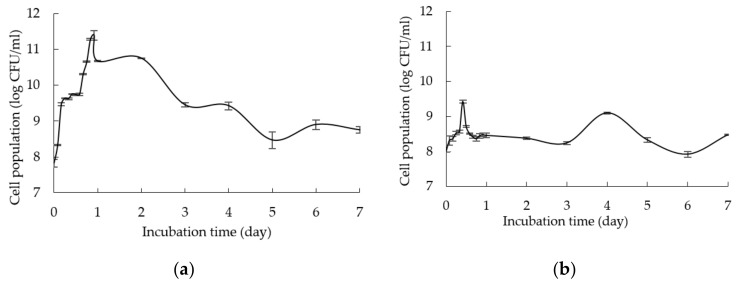
Growth profile of *Lactobacillus plantarum* RI 11 in: (**a**) control MRS medium (**b**) M1; rice straw and yeast extract (**c**) M2; rice straw and soybean pulp (**d**) M3; molasses and yeast extract (**e**) M4; molasses and soybean pulp (**f**) M5; PKC and yeast extract and (**g**) M6; PKC and soybean pulp. The error bars represent the standard deviation of cell population of triplicate samples (*n* = 3).

**Figure 2 molecules-25-02607-f002:**
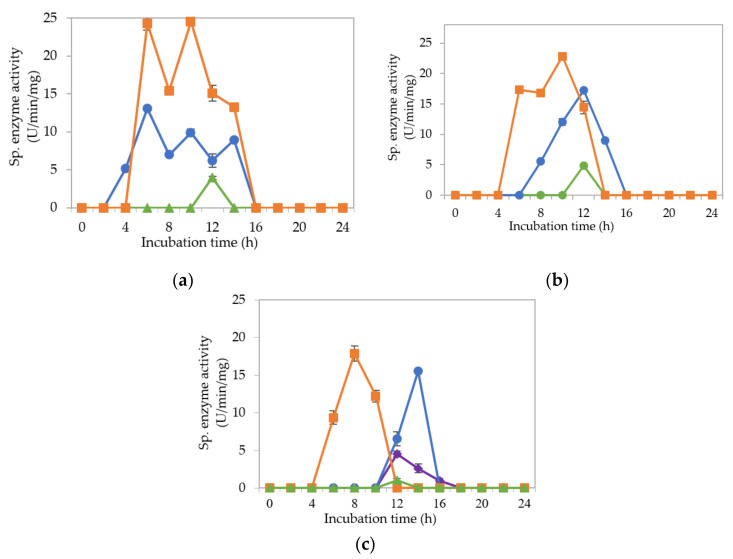
Specific extracellular cellulolytic and hemicellulolytic activities; (●) endoglucanase; (■) exoglucanase; (▲) β-glucosidase and (♦) mannanase of *Lactobacillus plantarum* RI 11 grown in MRS control medium for 24 h. The enzyme activities were detected at three different pH levels: (**a**) pH 5; (**b**) pH 6.5; (**c**) pH 8. The error bars represent the standard deviation of specific enzyme activity of triplicate samples (*n* = 3).

**Figure 3 molecules-25-02607-f003:**
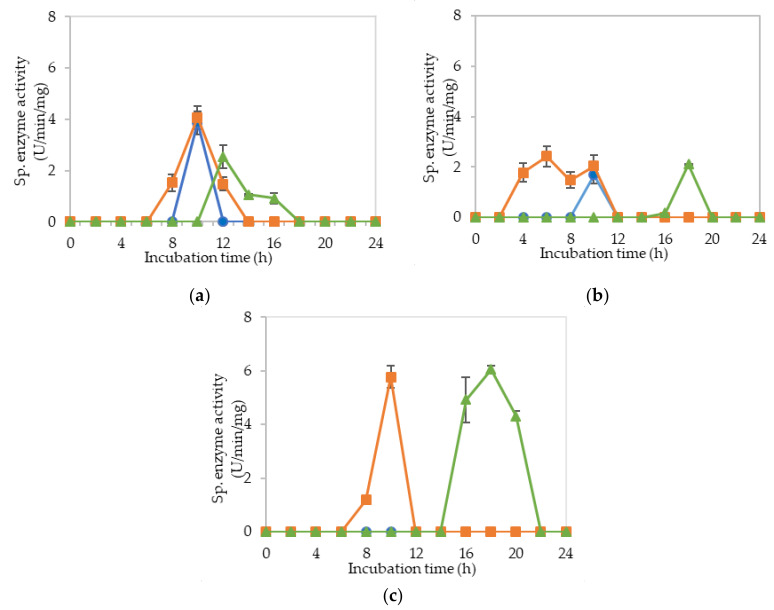
Specific extracellular cellulolytic and hemicellulolytic activities; (●) endoglucanase; (■) exoglucanase; (▲) β-glucosidase and (♦) mannanase of Lactobacillus plantarum RI 11 grown in M1 (rice straw and yeast extract) for 24 h. The enzyme activities were detected at three different pH levels: (**a**) pH 5; (**b**) pH 6.5; (**c**) pH 8. The error bars represent the standard deviation of specific enzyme activity of triplicate samples (*n* = 3).

**Figure 4 molecules-25-02607-f004:**
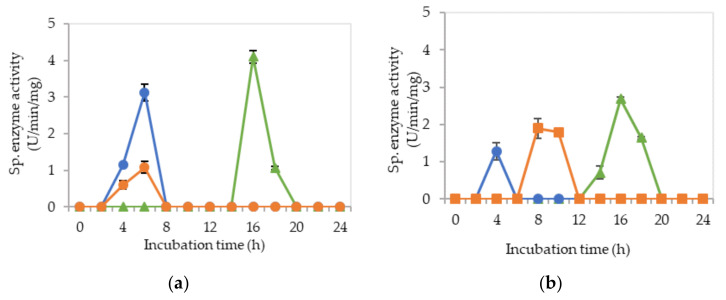
Specific extracellular cellulolytic and hemicellulolytic activities; (●) endoglucanase; (■) exoglucanase; (▲) β-glucosidase and (♦) mannanase of Lactobacillus plantarum RI 11 grown in M2 (rice straw and soybean pulp) for 24 h. The enzyme activities were detected at three different pH levels: (**a**) pH 5; (**b**) pH 6.5; (**c**) pH 8. The error bars represent the standard deviation of specific enzyme activity of triplicate samples (*n* = 3).

**Figure 5 molecules-25-02607-f005:**
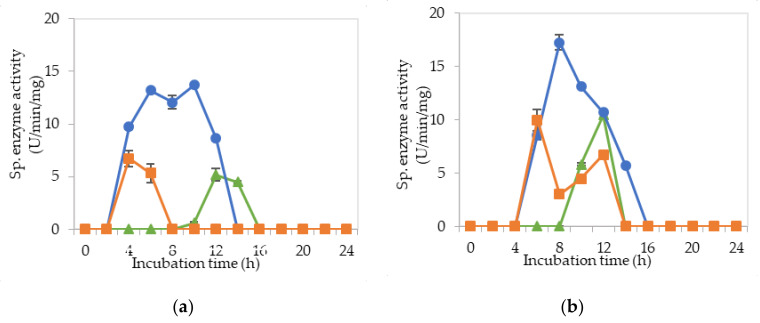
Specific extracellular cellulolytic and hemicellulolytic activities; (●) endoglucanase; (■) exoglucanase; (▲) β-glucosidase and (♦) mannanase of *Lactobacillus plantarum* RI 11 grown in M3 (molasses and yeast extract) for 24 h. The enzyme activities were detected at three different pH levels: (**a**) pH 5; (**b**) pH 6.5; (**c**) pH 8. The error bars represent the standard deviation of specific enzyme activity of triplicate samples (*n* = 3).

**Figure 6 molecules-25-02607-f006:**
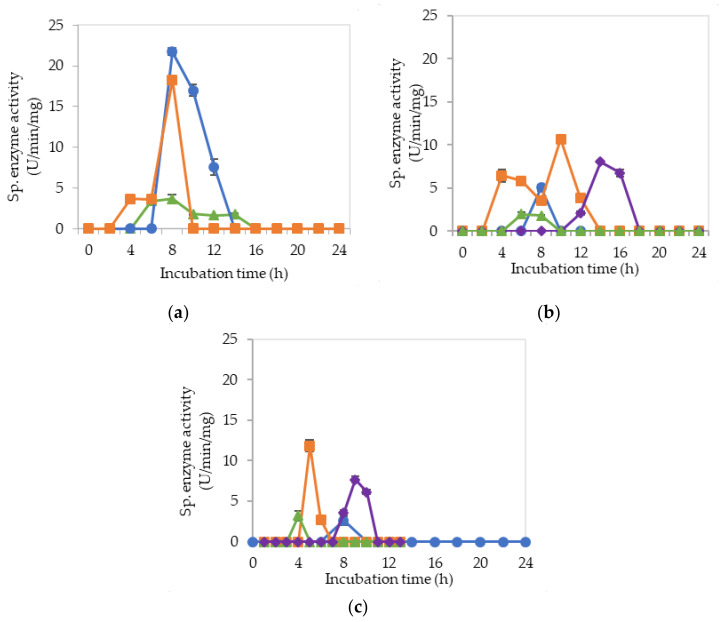
Specific extracellular cellulolytic and hemicellulolytic activities; (●) endoglucanase; (■) exoglucanase; (▲) β-glucosidase and (♦) mannanase of *Lactobacillus plantarum* RI 11 grown in M4 (molasses and soybean pulp) for 24 h. The enzyme activities were detected at three different pH levels: (**a**) pH 5; (**b**) pH 6.5; (**c**) pH 8. The error bars represent the standard deviation of specific enzyme activity of triplicate samples (*n* = 3).

**Figure 7 molecules-25-02607-f007:**
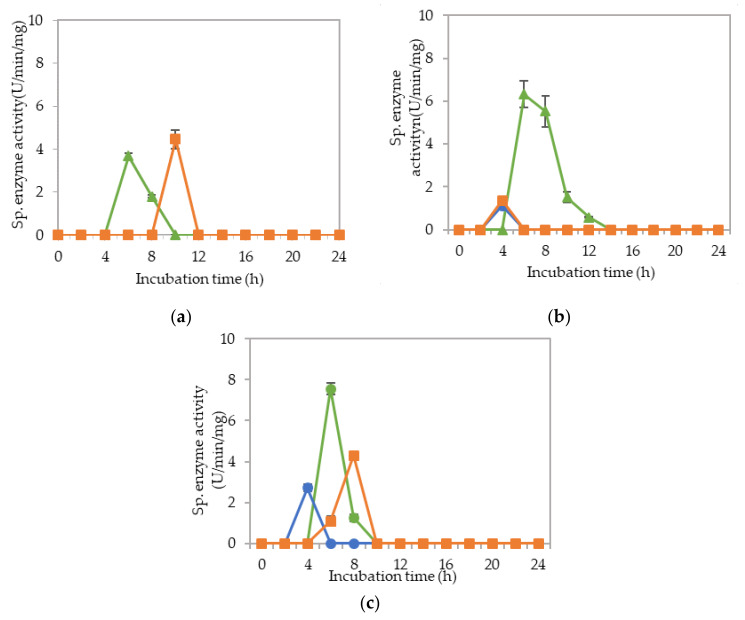
The specific extracellular cellulolytic and hemicellulolytic activities; (●) endoglucanase; (■) exoglucanase; (▲) β-glucosidase and (♦) mannanase of *Lactobacillus plantarum* RI 11 grown in M5 (PKC and yeast extract) for 24 h. The enzyme activities were detected at three different pH levels: (**a**) pH 5; (**b**) pH 6.5; (**c**) pH 8. The error bars represent the standard deviation of specific enzyme activity of triplicate samples (*n* = 3).

**Figure 8 molecules-25-02607-f008:**
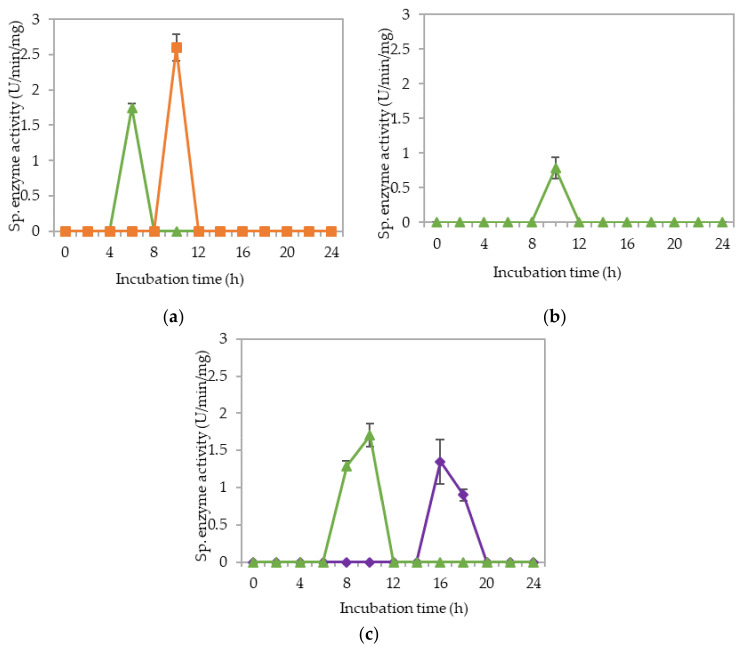
Specific extracellular cellulolytic and hemicellulolytic activities; (●) endoglucanase; (■) exoglucanase; (▲) β-glucosidase and (♦) mannanase of Lactobacillus plantarum RI 11 grown in M6 (PKC and soybean pulp) for 24 h. The enzyme activities were detected at three different pH levels: (**a**) pH 5; (**b**) pH 6.5; **(c**) pH 8. The error bars represent the standard deviation of specific enzyme activity of triplicate samples (*n* = 3).

**Figure 9 molecules-25-02607-f009:**
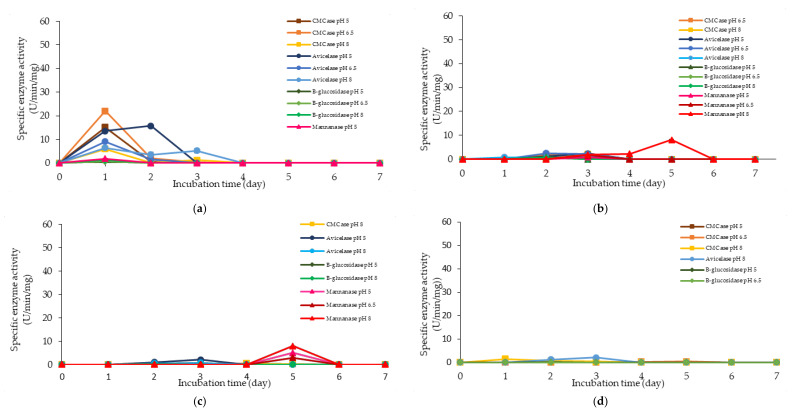
Specific extracellular cellulolytic and hemicellulolytic activities of Lactobacillus plantarum RI 11 grown in (**a**) F2; (**b**) F3; (**c**) F4; (**d**) F5; (**e**) F6; (**f**) F7; (**g**) F8; (**h**) F9; (**i**) F10; (**j**) F11; (**k**) F12; (**l**) F13; (**m**) F14; (**n**) F15; (**o**) F16; (**p**) MRS; (**q**) CRMRS during seven days of incubation. The enzyme activities were detected at three different pH levels. The error bars represent the standard deviation of specific enzyme activity of triplicate samples (*n* = 3). The medium composition was listed in Table 4.

**Table 1 molecules-25-02607-t001:** Viable cell count of *Lactobacillus plantarum* RI 11 grown in media designed by FFD.

Day	Viable Cell Density(Log CFU/mL)								
	F1	F2	F3	F4	F5	F6	F7	F8	F9
**0**	7.99 ± 0.04^Ad^	8.53 ± 0.00^Db^	9.43 ± 0.21^Ba^	9.25 ± 0.29^Ea^	9.40 ± 0.19^Da^	9.25 ± 0.15^Fa^	8.48 ± 0.04^Cbc^	8.28 ± 0.03^Dbcd^	7.91 ± 0.10^Dd^
**1**	7.85 ± 0.30^Ae^	9.62 ± 0.26^Ee^	9.70 ± 0.04^Ba^	9.74 ± 0.03^Da^	9.63 ± 0.11^Dab^	9.67 ± 0.04^DEa^	8.43 ± 0.02^Cd^	9.50 ± 0.04^Aab^	9.56 ± 0.09^Cab^
**2**	6.56 ± 0.14^Bj^	9.73 ± 0.03^Bd^	9.73 ± 0.02^Bd^	9.73 ± 0.02^Dd^	9.64 ± 0.04^Dd^	9.76 ± 0.01^Dd^	7.57 ± 0.08^Di^	9.22 ± 0.04^Be^	10.50 ± 0.01^Ac^
**3**	6.40 ± 0.10^Bc^	10.05 ± 0.07^Bc^	10.14 ± 0.09^Aab^	10.45 ± 0.03^BCab^	10.46 ± 0.05^Cab^	10.73 ± 0.01^Ba^	9.05 ± 0.19^Babc^	8.13 ± 0.14^DEabc^	9.41 ± 0.12^Cab^
**4**	5.92 ± 0.09^Ck^	10.61 ± 0.23^Ac^	9.20 ± 0.01^Cef^	11.76 ± 0.01^Aa^	11.76 ± 0.01^Aa^	10.99 ± 0.05^Bb^	8.68 ± 0.02^Chi^	8.74 ± 0.10^Cghi^	9.32 ± 0.07^Ce^
**5**	5.30 ± 0.00^Dj^	9.25 ± 0.18^Cg^	10.42 ± 0.09^Ad^	10.74 ± 0.01^Bc^	11.69 ± 0.04^Aa^	11.43 ± 0.12^Ab^	9.49 ± 0.09^Agf^	9.43 ± 0.04^ABgf^	10.54 ± 0.07^Acd^
**6**	0 ± 0^Ej^	9.77 ± 0.08^Bc^	9.26 ± 0.16^Cde^	10.34 ± 0.06^BCb^	11.10 ± 0.07^Da^	9.43 ± 0.13^EFd^	7.46 ± 0.06^Di^	7.94 ± 0.11^Eh^	10.42 ± 0.06^Ab^
**7**	0 ± 0^Eh^	10.20 ± 0.17^ABb^	10.19 ± 0.16^Ab^	10.14 ± 0.16^Dbc^	11.29 ± 0.07^Ba^	10.05 ± 0.16^Cbc^	7.28 ± 0.17^Dg^	7.90 ± 0.10^Ef^	10.01 ± 0.11^Bbc^
	**F10**	**F11**	**F12**	**F13**	**F14**	**F15**	**F16**	**MRS**	**CRMRS**
**0**	8.08 ± 0.17^Dcd^	8.28 ± 0.04^BCbcd^	8.12 ± 0.14^CDbcd^	8.20 ± 0.01^Dbcd^	8.13 ± 0.12^Cbcd^	8.29 ± 0.16^Ebcd^	8.19 ± 0.05^Ebcd^	9.13 ± 0.03^Ea^	9.01 ± 0.19^Ca^
**1**	9.67 ± 0.08^Aa^	9.29 ± 0.08^Abc^	7.89 ± 0.08^De^	9.04 ± 0.09^Ac^	8.23 ± 0.10^Cd^	9.69 ± 0.04^Ca^	9.73 ± 0.02^Ca^	9.65 ± 0.00^Dab^	9.62 ± 0.04^Bab^
**2**	9.68 ± 0.06^Ad^	9.14 ± 0.02^Ae^	7.87 ± 0.17^Dh^	9.00 ± 0.02^Af^	8.26 ± 0.08^Cg^	10.76 ± 0.01^Aa^	10.56 ± 0.02^Ac^	10.73 ± 0.01^Aab^	9.70 ± 0.02^ABd^
**3**	8.45 ± 0.11^Cabc^	8.07 ± 0.03^Cabc^	7.86 ± 0.05^Dbc^	8.44 ± 0.07^Cabc^	8.24 ± 0.02^Cabc^	10.76 ± 0.01^Aa^	8.42 ± 0.05^Eabc^	10.10 ± 0.08^Cab^	9.99 ± 0.16^Aab^
**4**	9.15 ± 0.02^Bef^	8.21 ± 0.08^BCj^	8.52 ± 0.16^ABi^	8.60 ± 0.05^BCi^	8.59 ± 0.14^Bi^	9.99 ± 0.08^Bd^	8.92 ± 0.18^Dfgh^	10.36 ± 0.05^Bc^	8.99 ± 0.07^Cfg^
**5**	9.40 ± 0.08^Bgf^	8.36 ± 0.09^Bi^	8.73 ± 0.02^Ah^	8.77 ± 0.01^Bh^	8.32 ± 0.04^Ci^	10.16 ± 0.08^Be^	9.97 ± 0.05^BCe^	10.62 ± 0.08^Acd^	9.50 ± 0.08^Bf^
**6**	9.31 ± 0.06^Bde^	9.11 ± 0.08^Aef^	8.40 ± 0.10^ABCg^	8.20 ± 0.06^Dgh^	8.95 ± 0.15^Af^	9.31 ± 0.07^Dde^	10.16 ± 0.11^Bb^	9.43 ± 0.05^Dd^	8.46 ± 0.06^Dg^
**7**	9.30 ± 0.01^Bd^	7.66 ± 0.14^Df^	8.35 ± 0.07^BCe^	7.89 ± 0.10^Ef^	9.08 ± 0.11^Ad^	9.36 ± 0.06^Dd^	9.78 ± 0.08^Cc^	9.04 ± 0.20^Ed^	8.54 ± 0.04^De^

Note: Mean ± SEM. The same superscript within the same column (lowercase) indicates not significantly different (*p* < 0.05). Mean ± SEM. The same superscript within the same row (uppercase) indicates not significantly different (*p* < 0.05).

**Table 2 molecules-25-02607-t002:** Carbon and nitrogen sources supplemented to each formulated medium.

Medium	C Source	Concentration (g/L)	N Source	Concentration (g/L)
M1	Rice straw	15.46	Yeast extract	28.34
M2	Rice straw	15.46	Soybean pulp	51.54
M3	PKC	11.86	Yeast extract	28.34
M4	PKC	11.86	Soybean pulp	51.54
M5	Molasses	25.09	Yeast extract	28.34
M6	Molasses	25.09	Soybean pulp	51.54
Control	Glucose	20.00	Yeast extract	28.34

**Table 3 molecules-25-02607-t003:** Carbon and nitrogen contents for renewable natural polymers employed in this study.

	Component	Carbon Content (%)	Nitrogen Content (%)
Renewable natural polymers	PKC	31.00	-
	Molasses	14.65	-
	Rice straw	23.77	-
	Soybean pulp	-	35.77
Reconstituted MRS medium	Glucose	18.38	-
	Yeast extract	-	65.05

**Table 4 molecules-25-02607-t004:** FFD matrix for six variables with coded values for the growth enhancement, cellulolytic and hemicellulolytic enzyme productions of *Lactobacillus plantarum* RI 11.

Run	Factor					
	A	B	C	D	E	F
F1	−1	−1	−1	−1	−1	−1
F2	1	−1	−1	−1	1	−1
F3	−1	1	−1	−1	1	1
F4	1	1	−1	−1	−1	1
F5	−1	−1	1	−1	1	1
F6	1	−1	1	−1	−1	1
F7	−1	1	1	−1	−1	−1
F8	1	1	1	−1	1	−1
F9	−1	−1	−1	1	−1	1
F10	1	−1	−1	1	1	1
F11	−1	1	−1	1	1	−1
F12	1	1	−1	1	−1	−1
F13	−1	−1	1	1	1	−1
F14	1	−1	1	1	−1	−1
F15	−1	1	1	1	−1	1
F16	1	1	1	1	1	1

MRS and CRMRS was used as a control. Factor A, Glucose (20.00 g/L); B, Molasses (25.09 g/L); C, Rice straw (15.46 g/L); D, PKC (11.86 g/L); E, Yeast extract (28.34 g/L); F, Soybean pulp (51.54 g/L).

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
