# Peer review of "Enhancement of Versatile Extracellular Cellulolytic and Hemicellulolytic Enzyme Productions by Lactobacillus plantarum RI 11 Isolated from Malaysian Food Using Renewable Natural Polymers"

_molecules, 2020, doi:10.3390/molecules25112607_

Round 1

Reviewer 1 Report

Dear Authors,

A few minor points might need to be addressed.

1) Keywords (Lactobacillus plantarum RI 11) might change to (Lactobacillus plantarum).

2) In the Figure 1 (a-g), Authors might consider to change the hour to day. Because 144 hours is long and difficult to get an immediate idea for the length of incubation period. Moreover, the x-axis values can be easily converted to number of days without adding additional line on x-axis.  

3) There are some inconsistent throughout the manuscript, for example Page 6, Line 212, between 4-16 h of incubation, but Page 6, Line 218, Figure 2c at 12 h and 14 h. Aside from those, the other parameter such as Page 6, Line 215, pH range of pH 5, pH 6.5 and pH 8, while Page 6, Line 221, was detected at pH 5, 6.5 and 8.

4) Authors might need to recheck the references to ensure the format was appropriate.

Example, (i) Ref 1, "Appl. Environ. Microbiol 1971" should be "Appl. Environ. Microbiol. 1971", (ii) Ref 3, "Journal of Applied Bacteriology" should be in abbreviation form, (iii) Ref 50, "Journal of bacteriology 1998, 180, 1750-1758" should be "Journal of bacteriology (use Journal abbreviated form) 1998, 180, 1750-1758", and others.

Author Response

Responses to Reviewer 1 Comments:

A few minor points might need to be addressed.

Point 1: Keywords (Lactobacillus plantarum RI 11) might change to (Lactobacillus plantarum).

Response 1:  

We have changed the Keywords of Lactobacillus plantarum RI 11 to Lactobacillus plantarum as suggested.

Point 2: In the Figure 1 (a-g), Authors might consider changing the hour to day. Because 144 hours is long and difficult to get an immediate idea for the length of incubation period. Moreover, the x-axis values can be easily converted to number of days without adding additional line on x-axis.

Response 2:

Thanks for the good suggestion! We have changed the x-axis values of Figure 1 (a-g) from hour to day accordingly.

Point 3: There are some inconsistent throughout the manuscript, for example Page 6, Line 212, between 4-16 h of incubation, but Page 6, Line 218, Figure 2c at 12 hand 14 h. Aside from those, the other parameter such as Page 6, Line 215, pH range of pH 5, pH 6.5 and pH 8, while Page 6, Line 221, was detected at pH 5, 6.5 and 8.

Response 3:

We have corrected the inconsistency throughout the manuscript accordingly.

Point 4: Authors might need to recheck the references to ensure the format was appropriate.

Example, (i) Ref 1, "Appl. Environ. Microbiol 1971" should be "Appl. Environ. Microbiol.1971", (ii) Ref 3, "Journal of Applied Bacteriology" should be in abbreviation form, (iii) Ref 50, "Journal of bacteriology 1998, 180, 1750-1758" should be "Journal of bacteriology (use Journal abbreviated form) 1998,180, 1750-1758", and others.

Response 4:

The references and citation were corrected accordingly at Line 863-1113.

Reviewer 2 Report

Jan 1, 2020.

Journal: molecules

ID: molecules-682414-peer-review-v1

Title: Enhancement of Versatile Extracellular Cellulolytic and Hemicellulolytic Enzyme Productions by Lactobacillus plantarum RI 11 Isolated from Malaysian Food Using Renewable Natural Polymers

Authors:  Nursyafiqah Athirah Mohamad Zabidi, Hooi Ling Foo, Teck Chwen Loh, Rosfarizan Mohamad, and Raha Abdul Rahim

Comments to Authors:

Authors list and affiliations The name of last author should be preceded by (and). The mid name for each author should be just the initial, i.e. (Nursyafiqah A. M. Zabidi) The email for each author should be added

Abstract Please improve the abstract Could you add the novely to the abstract The aim of the study is unclear The authors should add the graphic abstract 1, L.39 “supplemented that supplemented”, please check The abstract shouldn`t exceed 200 words. In the keywords: “carbon source” and “nitrogen source”, the author should use more reasonable and leading keywords.

Introduction 3, L.105 what do you mean by “Generally Regarded as Safe”? “LAB has not been reported as the biomass bioconversion agent elsewhere” re-consider this statement and please check reference. doi: 10.1186/s12934-014-0097-0. The authors would give detailes about “Fractional Factorial Design (FFD)” in the introduction The author should support the information of lines 97 to 102 with some reported references. Reference: “doi: 10.3390/ijms20204979” has the same idea so please explain the novelty of your study? Please add the aim of study at the end of introduction. Please refer to the previous studies. Could you mention the importance of this idea in industrial applications? The introduction is unfocused; repeated words, and poorly written

Results and Discussion Results and discussion should be separated according to the instructions of authors “Different renewable natural polymers”; please give examples What are the rationale for choosing the polymers in your study? Please explain the curves in the figures (1 a-g)? In fig 1. The author should uniform the text font, 2 figures has the same legend (b) and (c) is missing “endoglucanase, exoglucanase and β-glucosidase are vital to react synergistically in the mechanism of biomass degradation” please explain in more detailes “endoglucanase-CMCase, exoglucanase-avicelase and, β-glucosidase) activities, which could be detected”; the authors could mention how they detected these enzymes?” Results is very long and need to be rewritten in more proper and organized way There are a lot of abbreviation, please mention what are these abbreviations stand for? Please add the discussion and refer to the previous studies that support your results

Material and methods Could you add the amount of bacteria utilized in the experiments Please add the voucher code for the sample Each experiment should be triplicated The mount of natural polymers should be added

In the text, reference numbers should be placed in square brackets [ ] The references should be revised carefully based on the journal instructions:

Author 1, A.B.; Author 2, C.D. Title of the article. Abbreviated Journal Name. Year, Volume, page range, DOI or other identifier. Available online: URL (accessed on Day Month Year).

Taken together:

There are many grammatical errors and, some sentences don't make any sense. Many unclear or confusing parts The English style of the manuscript is very poor in many parts and hence the manuscript needs a comprehensive improvement of the English

The authors could benefit from the following reference in the introduction section:

El-Seedi, H.R., El-Said, A.M., Khalifa, S.A., Goransson, U., Bohlin, L., Borg-Karlson, A.K. and Verpoorte, R., 2012. Biosynthesis, natural sources, dietary intake, pharmacokinetic properties, and biological activities of hydroxycinnamic acids. Journal of Agricultural and Food chemistry, 60(44), pp.10877-10895.

Author Response

Response to Reviewer 2 Comments:

Point 1: Authors list and affiliations. The name of last author should be preceded by (and). The mid name for each author should be just the initial, i.e. (Nursyafiqah A. M. Zabidi). The email for each author should be added

Response 1:

The “Authors list and affiliations have been corrected accordingly together with the email for each author at Line 6-23.

Point 2: Abstract Please improve the abstract Could you add the novelty to the abstract The aim of the study is unclear The authors should add the graphic abstract 1, L.39 “supplemented that supplemented”, please check The abstract shouldn`t exceed 200 words. In the keywords: “carbon source” and “nitrogen source”, the author should use more reasonable and leading keywords.

Response 2:

We had revised the abstract accordingly by including the novelty and objectives of the study. The abstract was further improved and summarised, which is not exceeding 200 words by following the instruction of authors at Line 25-41. The graphic abstract was also added at Line 44.

Point 3: Introduction:

a) what do you mean by “Generally Regarded as Safe”?

b) “LAB has not been reported as the biomass bioconversion agent elsewhere” re-consider this statement and please check reference. doi: 10.1186/s12934-014-0097-0.

c) The authors would give details about “Fractional Factorial Design (FFD)” in the introduction.

d) The author should support the information of lines 97 to 102 with some reported references.

e) Reference: “doi: 10.3390/ijms20204979” has the same idea so please explain the novelty of your study? Please add the aim of study at the end of introduction. Please refer to the previous studies. Could you mention the importance of this idea in industrial applications?

f) The introduction is unfocused; repeated words, and poorly written.

Response 3:

Introduction:

a) We have explained the “Generally Regarded as Safe” in the Introduction section at Line 83- 88.

b) We have revised the sentence of “LAB has not been reported as the biomass bioconversion agent elsewhere” by including the following reference: doi: 10.1186/s12934-014-0097-0 at Line 103-119.

c) We have described briefly “Fractional Factorial Design in Introduction section at Line 52-56 and in “Results and discussion” at Line 500-504 respectively.

d) We have supported the information of lines 97 to 102 with some reported references at Lines 74-79 in the revised manuscript.

e) We have included the novelty, aims and the importance of this study, as well as its industrial application in the Introduction section at Line 103-119, whereby the Reference: “doi: 10.3390/ijms20204979” has been shown to have different idea to our present study/report.

f) We have revised and improved the overall introduction extensively at Line 46-119.

Point 4: Results and Discussion:

  1. Results and Discussion should be separated according to the instructions of authors.
  2. “Different renewable natural polymers”; please give examples What are the rationale for choosing the polymers in your study?
  3. Please explain the curves in the figures (1 a-g)? In fig 1. The author should uniform the text font, 2 figures has the same legend (b) and (c) is missing.
  4. “endoglucanase, exoglucanase and β-glucosidase are vital to react synergistically in the mechanism of biomass degradation” please explain in more details.
  5. “endoglucanase-CMCase, exoglucanase-avicelase and, β-glucosidase) activities, which could be detected”; the authors could mention how they detected these enzymes?”
  6. Results is very long and need to be rewritten in more proper and organized way.
  7. There are a lot of abbreviation, please mention what are these abbreviations stand for?
  8. Please add the discussion and refer to the previous studies that support your results

Response 4:

 1.The main reason to combine both results and discussion in the current manuscript is to facilitate the understanding of the results and discussion of the readers and to avoid repetition of result descriptions in the Discussion section if the results are presented separately from the Discussion section. Moreover, a few articles published in the “Molecules” journal have combined both results and discussion together, including our recent article published on 11 February 2020, doi:10.3390/molecules25040779.

2. We have included the examples for “Different renewable natural polymers” and the rationale for choosing the selected polymers in our study at Line 123-131.

3. We have explained and discussed the curves in the figure (1 a-g) accordingly at Line 133-220. We have also revised the text font of Figure 1 and corrected the legend (b) and (c) accordingly.

4. We have explained why endoglucanase, exoglucanase and β-glucosidase are vital to react synergistically in the mechanism of biomass degradation at Line 237-245.

5. We have described how “endoglucanase-CMCase, exoglucanase-avicelase and, β-glucosidase) activities we detected” at Line 245-252.

6. We have summarized the “Results” accordingly and rewritten in more proper and organized way, whereby we have discussed the results in such a way that similar trends of results were grouped together for the ease of discussion.

7. We have described all abbreviations in the results and discussion accordingly and the description of all abbreviations are also listed in the list of “Abbreviation” at Line 858-859.

8. The appropriate discussion was added after the respective results accordingly and supported by previous studies accordingly throughout the “Results and discussion” section.

Point 5: Material and methods:

  1. Could you add the amount of bacteria utilized in the experiments?
  2. Please add the voucher code for the sample
  3. Each experiment should be triplicated.
  4. The amount of natural polymers should be added.

 Response 5:

1. We have included the bacteria population used in the experiment at Line 724-725.

2. We have addressed all samples according to the Instruction to Authors.

3. We have conducted each experiment in triplicates as described in each experiment.

4. We have included the amount of natural polymers used at Line 731-734 and Line 807-809 respectively.

Point 6: In the text, reference numbers should be placed in square brackets [ ]. The references should be revised carefully based on the journal instructions:

Response 6:

The citations of references in the text were corrected accordingly and the references were revised according to the journal instructions.

Point 7: Author 1, A.B.; Author 2, C.D. Title of the article. Abbreviated Journal Name. Year, Volume, page range, DOI or other identifier. Available online: URL (accessed on Day Month Year).

Response 7:

The reference list was corrected according to the standard format from Line 863-1113.

Point 8: There are many grammatical errors and, some sentences don't make any sense. Many unclear or confusing parts. The English style of the manuscript is very poor in many parts and hence the manuscript needs a comprehensive improvement of the English.

Response 8:

We have revised and improved the overall manuscript accordingly. The manuscript has been proofread by an English language professional.

Point 9: The authors could benefit from the following reference in the introduction section:

El-Seedi, H.R., El-Said, A.M., Khalifa, S.A., Goransson, U., Bohlin, L., Borg-Karlson, A.K. and Verpoorte, R., 2012. Biosynthesis, natural sources, dietary intake, pharmacokinetic properties, and biological activities of hydroxycinnamic acids. Journal of Agricultural and Food chemistry, 60(44), pp.10877-10895.

Response 9:

Thanks for the suggestion! However, the objective of this study was to evaluate the effects of various renewable natural polymers on the growth and production of extracellular cellulolytic and hemicellulolytic enzymes by the novel isolate of L. plantarum RI 11. Hence, we wish to consider the suggested article as a reference in future if related.

Round 2

Reviewer 2 Report

Dear Editor

Yes, it has been modified according to our suggestions.

I recommend the paper for publication.

Kindest regards, Hesham